# Alcohol Withdrawal Is an Oxidative Stress Challenge for the Brain: Does It Pave the Way toward Severe Alcohol-Related Cognitive Impairment?

**DOI:** 10.3390/antiox11102078

**Published:** 2022-10-21

**Authors:** Virgile Clergue-Duval, Laurent Coulbault, Frank Questel, Nicolas Cabé, Alice Laniepce, Clément Delage, Céline Boudehent, Vanessa Bloch, Shailendra Segobin, Mickael Naassila, Anne-Lise Pitel, Florence Vorspan

**Affiliations:** 1Département de Psychiatrie et de Médecine Addictologique, Site Lariboisière Fernand-Widal, GHU APHP Nord–Université Paris Cité, APHP, F-75010 Paris, France; 2Inserm UMRS-1144 Optimisation Thérapeutique en Neuropsychopharmacologie, Université Paris Cité, F-75006 Paris, France; 3FHU Network of Research in Substance Use Disorders (NOR-SUD), F-75006 Paris, France; 4Resalcog (Réseau Pour la Prise en Charge Des Troubles Cognitifs Liés à L’alcool), F-75017 Paris, France; 5Normandie Univ, UNICAEN, INSERM, U1237, PhIND “Physiopathology and Imaging of Neurological Disorders”, Institut Blood and Brain @ Caen-Normandie, Cyceron, F-14074 Caen, France; 6FHU Améliorer le Pronostic Des Troubles Addictifs et Mentaux Par Une Médecine Personnalisée (A2M2P), F-14074 Caen, France; 7Service d’Addictologie, Centre Hospitalier Universitaire de Caen, F-14000 Caen, France; 8Normandie Univ, UNIROUEN, CRFDP (EA 7475), Rouen F-76000, France; 9Service de Pharmacie, Site Lariboisière Fernand-Widal, GHU APHP Nord–Université Paris Cité, APHP, F-75010 Paris, France; 10UFR de Pharmacie, Université Paris Cité, F-75006 Paris, France; 11Normandie Univ, UNICAEN, PSL Université Paris Cité, EPHE, INSERM, U1077, CHU de Caen, GIP Cyceron, Neuropsychologie et Imagerie de la Mémoire Humaine, F-14074 Caen, France; 12Inserm UMRS-1247 Groupe de Recherche Sur L’alcool et Les Pharmacodépendances, Université de Picardie Jules Verne, F-80000 Amiens, France; 13UFR de Pharmacie, Université de Picardie Jules Verne, F-80000 Amiens, France; 14UFR de Médecine, Université Paris Cité, F-75006 Paris, France

**Keywords:** oxidative stress, alcohol withdrawal, alcohol use disorder, alcohol brain damage, Wernicke encephalopathy, cognitive impairment, thiamine, diathesis

## Abstract

Alcohol use is a leading cause of mortality, brain morbidity, neurological complications and minor to major neurocognitive disorders. Alcohol-related neurocognitive disorders are consecutive to the direct effect of chronic and excessive alcohol use, but not only. Indeed, patients with severe alcohol use disorders (AUD) associated with pharmacological dependence suffer from repetitive events of alcohol withdrawal (AW). If those AW are not managed by adequate medical and pharmacological treatment, they may evolve into severe AW, or be complicated by epileptic seizure or *delirium tremens* (DT). In addition, we suggest that AW favors the occurrence of Wernicke’s encephalopathy (WE) in patients with known or unknown thiamine depletion. We reviewed the literature on oxidative stress as a core mechanism in brain suffering linked with those conditions: AW, epileptic seizure, DT and WE. Thus, we propose perspectives to further develop research projects aiming at better identifying oxidative stress brain damage related to AW, assessing the effect of repetitive episodes of AW, and their long-term cognitive consequences. This research field should develop neuroprotective strategies during AW itself or during the periwithdrawal period. This could contribute to the prevention of severe alcohol-related brain damage and cognitive impairments.

## 1. Introduction

Alcohol use is a major preventable risk factor for human health. The global health burden initiative estimated that any alcohol use is associated with an increased risk [1], except from zero drink per week, which is the only no-risk consumption. Increased alcohol use was associated with 2.2% of all age-standardised female deaths and 6.8% of all age-standardised male deaths. In the younger age groups (15–49 years), alcohol use is associated with specific death causes (tuberculosis, road accidents, self-harm), and those causes of deaths change in older age groups, with cancer taking the leading rank after 50 years of age [1]. When trying to estimate alcohol-use related premature mortality (below 69 years of age) worldwide, Sohi et al. identified cirrhosis of the liver (457,000 deaths), road injuries (338,000 deaths), and tuberculosis (190,000 deaths) as the leading causes of the burden of alcohol-attributable premature deaths [2]. In a country such as France, where alcohol production and consumption is regarded as part of the culture, alcohol-related mortality is estimated to be 40,000 deaths per year [3], representing 13% of all causes for males and 5% for females. In the various causes of alcohol-attributed deaths, a specific category can be specified in death certificates including “degeneration of nervous system due to alcohol” or “Wernicke’s encephalopathy” (WE), “mental and behavioural disorders due to alcohol” or other alcohol-related brain diseases such as “epilepsy and status epilepticus” or “stroke”.

### 1.1. Alcohol-Induced Brain Morbidity

Apart from mortality, alcohol use is also associated with a high worldwide burden in terms of morbidity: various forms of cancers, liver cirrhosis, and brain disorders [4]. Those alcohol-induced brain disorders include psychiatric diseases such as Alcohol Use Disorder (AUD), self-harm, interpersonal violence, acute neurological conditions such as epilepsy or haemorrhagic stroke, but also chronic cognitive deficits including Korsakoff’s syndrome and various forms of dementia. A transient cognitive impairment is observable in 30 (for processing speed) to 50 (for episodic, working memory, inhibition capabilities) and up to 80% (for flexibility) of the patients examined early after a few days of inpatient detoxification and the cessation of benzodiazepine treatment [5]. These neuropsychological impairments can be observed in AUD patients’ history years before they meet the criteria for a persistent form of cognitive disorder. Moreover, in a nationwide French study, being diagnosed with “AUD” preceded 44% of all early-onset dementia diagnoses [6]. In that study, the diagnosis of AUD in any hospital-based file was the strongest modifiable risk factor for dementia onset, with an adjusted hazard ratio of 3.34 for women and 3.36 for men, while the hazard ratio never exceeded 2 for the other risk factors considered in this study.

### 1.2. Are All Heavy Alcohol Users at Risk of Brain Morbidity?

For mortality as well as for brain morbidity, the strong association with AUD does not imply that AUD is a specific risk factor. The same association could be observed in other forms of chronic exposure to large amounts of alcohol without the specific criteria of AUD as the result of a cumulative dose-effect relationship.

On the one hand, for the cumulative hypothesis, alcohol-related all-causes mortality studies have established twenty years ago that this mortality was mostly observable in subjects with the highest amount of alcohol use, especially in subjects reporting several days of heavy drinking per week [7].

But on the other hand, favoring a specific role of AUD, it has been established from population studies that among alcohol current users, subjects meeting the categorical DSM-IV definition of alcohol dependence reported higher scores to all alcohol use measures, including number of days with either excessive alcohol use, or heavy drinking or acute alcohol intoxications [8]. In addition, a more recent epidemiological study assessing alcohol-related excessive mortality has shown that this mortality is supported by subjects with alcohol dependence or severe AUD as defined by the presence of six or more criteria of the dimensional DSM-5 definition [9]. For the association between AUD and dementia [6], the association was measured with the specific condition of a hospital-based AUD diagnosis.

Therefore, our core hypothesis is that there is a specific cognitive toxicity supported by the condition of severe AUD, through the subgroup of patients with pharmacological dependence, as they experiment the repetition and/or the severity of acute episodes of alcohol withdrawal (AW). We searched the literature for the points supporting this hypothesis and putative mechanisms of this additional brain toxicity due to inadequately treated AW syndrome.

## 2. Alcohol Withdrawal: Definition

AUD is defined by the loss of control over alcohol intake and chronic, compulsive, heavy alcohol use despite adverse consequences [10]. Among patients seeking treatment for AUD, the proportion of patients at treatment entry endorsing the criteria for pharmacological dependence was 63% for tolerance and 14% for withdrawal [11]. AW syndrome is the combination of signs and symptoms occurring as soon as three to six hours after the last intake of alcohol in subjects with pharmacological dependence. The classical symptoms are tremor, perspiration, anxiety and adrenergic signs (hypertension, tachycardia). Untreated AW can lead to specific complications: *delirium tremens* (DT) and seizure. Several indirect complications of the adrenergic syndrome may also occur during an untreated AW syndrome as dehydration, cardiac failure [12] or renal failure. Mortality reaches 8% in patients with AW syndrome hospitalized in intensive care units, because of any or the combination of those multiple organs complications [13]. AW is still considered as a dangerous complication of undetected AUD during any surgery or medical inpatient treatment [14]. Milder forms of AW may also be observed during scheduled alcohol cessation, but it is easily preventable or treatable in ambulatory practice [15,16,17].

Recommended pharmacological treatment to lower the symptoms of AW and to avoid the associated complications can be divided in three categories: First, the treatment of withdrawal symptoms. Benzodiazepines are the first choice treatment [15,16,17], and should be adjusted to the clinical signs, sometimes with the help of scales to monitor the time course of those symptoms under treatment [18]. The adjunction of anti-adrenergic medications (alpha-2 agonist or beta-blockers) is recommended to control the hyperadrenergic state if insufficiently reduced by the benzodiazepines [16], but not recommended by all [15,17]. Second, the preventive treatments, to be taken even before the alcohol cessation is scheduled, or at least during the withdrawal itself. Even though regarding WE as a direct neurological complication of AW is controversial, it is usually recommended to include thiamine supplementation to avoid WE [15,17,19], especially in cases of known nutritional depletion. If the recommendation is clear when a withdrawal requires intensive care, the place of this supportive nutritional care in the treatment of every ambulatory and/or inpatient AW seems less imperative, and the recommendation is weaker [16,20]. The relationships between AW, thiamine deficiency and WE will be described later in the article. Third, for ambulatory mild withdrawal symptoms, some medical societies recommend, as a second line option, the use of GABAergic treatments not belonging to the benzodiazepines category, although their grade of evidence is weaker due to fewer studies [16,17]. These include gabapentine, carbamazepine or baclofen. All medical societies recommend considering AW, whether scheduled or accidental, as an opportunity to start a long-term treatment for AUD because it indicates a pharmacological dependence.

## 3. Alcohol Withdrawal as an Oxidative Stress Challenge for the Brain

As ethanol is a small molecule that easily passes the blood–brain barrier and is ubiquitously distributed in the brain, chronic alcohol use induces neuroadaptation [21,22]. This neuroadaptation is obviously revealed in cases of observable pharmacological dependence, characterized by the phenomenon of tolerance and the presence of signs of withdrawal when alcohol use ceases. Thus, at the brutal alcohol cessation, the equilibrium resulting from this neuroadaptation is disrupted and it will take several days to restore the balance. What occurs at the whole brain level is mainly an increase in glutamate and norepinephrine, a decrease in GABA and an increase in intracellular calcium concentration [21,23,24]. This hyperadrenergic state induces the physical signs of AW, but in actual practice, no direct clinical evaluation of the glutamatergic surge exists. As the result of this hyperglutamatergic state, a central nervous system hyperexcitability and an autonomic hyperactivity occurs. Then, a vicious cycle of AW exacerbating both central nervous system hyperexcitability and an autonomic hyperactivity may start. This leads to an intense oxidative stress and reactive oxygen species (ROS) production during early alcohol withdrawal [25,26], via the increase in excitatory neurotransmitters and intracellular calcium concentration [22]. At present, it is not known whether these changes in oxidative brain status persist after the initial period of AW [22]. Interestingly, the antioxidant N-acetylcysteine has been shown to be effective in reducing alcohol drinking and relapse in chronic alcohol consuming rats, and its efficacy may come from its ability to normalize glutamatergic homeostasis but also from its antioxidative properties [27,28,29]. However, the specific impact of this oxidative stress during AW has not been established per se [22], but it is a strong candidate mechanism that could contribute to the development of brain injuries and alcohol-related neurocognitive deficits.

### 3.1. Epilepsy, Delirium Tremens as AW Neurological Complications: The Role of Oxidative Stress

The two major and well-known complications of AW, occurring in case of absence or inadequate medical management of AW symptoms, are epileptic seizure and DT. Seizures are provoked by the hyperexcitability of neurons and intracellular calcium dysregulation [24], associated with neuroinflammation and an excessive production of ROS. They induce a synchronous depolarization of neurons, itself increasing the oxygen consumption by neurons and the production of ROS, maintaining and spreading the seizure [30]. The development of innovative treatment for epilepsy includes, amongst other pathways, molecules targeting the production of ROS [31]. DT, also known as AW delirium, is a complication of AW constituted by an acute confusion state in addition to the signs and symptoms of AW (tremor, perspiration and hyperadrenergic signs). It is a life-endangering condition requiring intensive care. Consequently, experimental studies in humans trying to link DT with oxidative stress are scarce. To our knowledge, one study assessed 8-hydroxy-2′-deoxyguanosine (8-OHdG), a peripheral blood biomarker of oxidative DNA damage, in patients with pharmacological dependence undergoing AW with and without DT [32]. The 8-OhdG level was higher in the DT group, but some patients of the AW without DT group also had elevated levels, suggesting the hyperexcitability of AW is sufficient to induce an oxidative stress. DT may not be categorically different from AW, but only dimensionally a more severe form of AW.

### 3.2. AW as a Vulnerability Period for Wernicke’s Encephalopathy: The Role of Oxidative Stress

WE is caused by the brain toxicity related to an acute thiamine diphosphate deficiency (also named thiamine pyrophosphate) or an altered thiamine metabolism and distribution [33,34,35,36,37,38,39]. Even though WE is not the direct consequence of AW only, this particular timeframe of “periwithdrawal” should be considered as a vulnerability period for the occurrence of WE.

#### 3.2.1. Wernicke’s Encephalopathy Occurrence

WE is frequent in AUD patients [39]. It is estimated that it affects between 10 and 35% of patients with AUD assessed when they are hospitalized for AW [39,40]; however, the prevalence is not known outside of the withdrawal period in AUD patients with active chronic alcohol use. In the general population, the prevalence of WE is much lower, and would be 0.4 to 2.8% [38]. As with the other complication of the thiamine deficiency *spectrum* [41], WE also appears in non-AUD patients during events of subacute (or more often acute) undernutrition, such as starvation, anorexia and food refusal, episodes of severe and repeated vomiting, malabsorption phenomena (exacerbation of inflammatory bowel disease or after bariatric surgery), but also in the context of hypercatabolism (neoplasia, sepsis, malaria, thyrotoxicosis) [42,43,44,45]. The prevalence of WE in those conditions is not precisely known, but the phenomenon is rare, and does not compare to the 10 and 35% of patients with AUD assessed when they are hospitalized for AW [39,40].

#### 3.2.2. Wernicke’s Encephalopathy Diagnosis

The clinical Caine’s criteria for the identification of WE are the presence of two criteria among: ataxia, oculomotor dysfunction, confusion and dietary deficiency [46]. However, the clinical triad of the neurologic symptoms (ataxia, oculomotor dysfunction and confusion) is often incomplete [36,38]. The criteria, especially when they are subtle, are often difficult to identify in chronic alcohol users because there may be ambiguity between WE and signs of acute alcohol intoxication, or in non-alcohol users, because of a lack of knowledge of this symptomatology and a poor assessment of the risk of malnutrition, leading to an underdiagnosis [36,38,40].

Regarding the imaging diagnosis of WE, brain magnetic resonance imaging (MRI) has a good specificity but a low sensitivity in clinical practice. It shows bilateral FLAIR hyperintensities in specific regions, such as the mammillary bodies, thalamus, hypothalamus, periaqueductal region and floor of the fourth ventricle [38]. In the context of this low sensitivity, it is actually the case that the value of MRI and imaging in clinical practice is mainly to rule out differential diagnoses [47].

Regarding blood assays, free thiamine (the inactive form) represents a small part of the total thiamine. It is rarely impaired and is not an indicator of thiamine status [37,48]. To objectify thiamine deficiency, the esters should be assessed (only thiamine diphosphate is assessed in clinical practice) by high-performance liquid chromatography or indirectly assessed by measuring the erythrocyte transketolase activity [37,48,49]. These assays are not performed in routine practice and are highly heterogeneous between laboratories [48]. In addition, the phosphorylation capacity and the active passage of the blood–brain barrier contribute to the cerebral availability of thiamine [37,50,51,52]. As a result, the value of thiamine blood measurement is relatively poor, and it is not recommended in clinical practice. Thiamine blood measurement strategies are not shown to be effective in comparison with systematic supplementation in AUD patients hospitalized for AW [53]. Because the diagnosis of WE in non-AUD patients can occur during various medical conditions leading to thiamine depletion, the blood quantification is relevant to provide an etiological diagnostic. Indeed, low blood concentration in free thiamine or thiamine diphosphate during a comprehensive confusion investigation or in case of suspicion of thiamine deficiency confirms the diagnosis [41].

#### 3.2.3. Systematic Thiamine Supplementation in AUD Patients

In clinical addiction medicine practice, given the prevalence of WE in hospitalized AUD patients, the absence of and the low sensitivity of the MRI or thiamine blood measurements, the majority of international medical societies recommend to systematically prescribe thiamine to patients displaying signs of AW [19]. Some recommendations advise prescribing according to the individual risk of WE notably based on the nutritional status, or the severity of the AW (delirium, intensity of withdrawal, signs of malnutrition, liver disease) [19]. Only the Australian recommendation indicates that thiamine needs to be continued indefinitely in AUD patients when alcohol consumption continues [15]. The other official recommendations do not refer to the appropriate management outside the period of acute AW [19], although thiamine prescription is a common practice.

#### 3.2.4. Wernicke’s Encephalopathy and Alcohol Withdrawal

Animal models of alcohol-related brain damage display similar brain and cognitive alterations to what is seen in AUD patients. Numerous studies have shown that thiamine deficiency [54], or even subclinical thiamine deficiency during chronic heavy alcohol consumption, is critical for the development of significant cognitive alterations affecting spatial memory and cognitive flexibility [55] associated with neuronal and neurotrophin loss.

However, only a few studies have investigated the interaction and the synergistic effects of both ethanol toxicity and thiamine deficiency [56,57,58] to induce brain lesions.

While clinical recommendations regarding the timing of preventive management and thiamine administration acknowledge a specific risk period during AW, the link between the physiological event of AW and the occurrence of WE is only suspected but not currently demonstrated [34,35].

In clinical practice, assessing the time course of WE as a consequence of AW or AW occurring in patients with subtle symptoms of WE is highly difficult. The difficulties include the absence of easily accessible plasma thiamine measurements [48,53], the frequent absence of any prior medical consultation enabling a comprehensive premorbid clinical evaluation, and the low sensitivity of imaging. Furthermore, the symptoms of WE and AW are difficult to disentangle. Here we will develop the pathophysiological data supporting our core hypothesis of AW favoring the occurrence of WE.

#### 3.2.5. Wernicke’s Encephalopathy and Oxidative Stress

The pathophysiology of WE is complex and multi-factorial, including glutamatergic excitotoxicity, oxidative stress, lactic acidosis and blood–brain barrier disruption [33] and cannot be restricted to the sole well-known thiamine deficiency.

Thiamine is an essential water-soluble vitamin required for the Krebs cycle and the production of adenosine triphosphate in the mitochondria [33,38]. Thiamine itself has antioxidant properties (Huang et al. 2010). Its deficiency leads to the depletion of adenosine triphosphate and the increase in lactates production [22,33,38].

During AW, by increasing catabolism, withdrawal symptoms could increase the consumption of thiamine in the Krebs cycle, deplete its reserves and induce a further increase in lactates. Thiamine is also needed for the pentose phosphate pathway and the production of nicotinamide adenine dinucleotide phosphate, which helps to eliminate ROS and to reduce lactic acidosis [33,38]. This role contributes to the increase in oxidative stress and the disruption of the homeostasis of cellular electrolytes [33,38]. In the brain, astrocytes are the main source of lactate production [33].

In addition, astrocytic dysfunction, induced by the thiamine deficiency and this increase in oxidative stress, alters its function of reuptake and metabolism of extracellular glutamate, especially with the loss of the glutamate transporters [33,59,60,61]. Using an animal model of Wernicke–Korsakoff syndrome, in which rats were submitted to a chronic ethanol treatment with or without a thiamine deficiency episode, the glutamate uptake was found to be reduced in the prefrontal cortex by thiamine deficiency, but not by chronic ethanol intake [62].

Confronted with this decreased capacity of metabolism by astrocytes, the hyperglutamatergia of AW could exacerbate the excitotoxicity caused by thiamine deficiency and worsen the induced neuronal deaths. This hyperglutamatergia contributes, notably in a context of calcium channels dysregulation by chronic alcohol use [24], to an increase in intracellular calcium concentration in neuronal and glial cells [21,61] and increases the production of ROS [22]. In practice, the hyperglutamatergic excitotoxicity is this major pathophysiological mechanism inducing the cerebral cells death and the histological lesions observed in WE [33]. Thus, we suggest a synergistic effect of the two excitotoxic phenomena, acute thiamine deficiency and AW that is summarized in Figure 1.

## 4. What Makes the Alcohol Withdrawal Period at High Risk for Oxidative Brain Damage?

AW occurs in subjects with severe AUD and pharmacological dependence. Those patients, by the combined effects of high chronic alcohol intake, poor diet, modified intestinal absorption and social disadvantages display a high rate of nutritional depletions [20,37,63,64,65]. Nutritional depletion puts them at high risk for brain suffering during the hyperglutamatergic and hyperadrenergic state induced by AW, potentially leading to WE in patients with subclinical thiamine deficiency or an individual genetic predisposition to thiamine deficiency [66]. Nutritional deficits include, apart from the previously discussed thiamine, magnesium and ascorbic acid (Vitamin C). Magnesium is notably a thiamine cofactor and also a glutamate NMDA receptor channel blocker. Beyond targeted correction in deficient patients, the relevance of its supplementation to reduce the intensity of AW and its consequences are discussed [67,68,69], although not recommended today in clinical practice. In addition, other nutritional interventions are discussed for the reduction in the oxidative stress and the prevention of neurotoxicity in AW, as ascorbic acid [63], nutritional ketosis [70] or omega-3 fatty acid treatments [71]. Those data suggest that a comprehensive nutritional assessment and/or supplementation should be developed for the peri-withdrawal period to prevent WE.

## 5. Alcohol Withdrawal and Severe Cognitive Impairments

We developed the hypothesis that AW cannot only result in severe neurological complications such as seizures or DT, but also be considered as a vulnerability period for the development of WE by exacerbating and aggravating its pathophysiological mechanisms. We now go beyond this hypothesis and suggest that even in AUD patients without WE and having not experienced seizures or DT during AW, the repetition or severity of AW could contribute to the development of cognitive deficits. Each AW episode represents an oxidative stress challenge for the brain, and may lead to a certain amount of neuronal, axonal, synaptic but also astrocytic destruction. Thus, a patient with severe AUD and pharmacological dependence may suffer from a cumulative effect of several such episodes. It is already known that repeated experience of AW results in a kindling-like process leading to increased likelihood and severity of epileptic seizures during detoxification [72]. One study has also observed that patients with a history of repeated detoxifications displayed more neurocognitive impairments than patients with a single, or no previous episode [73]. The severity of AW was also shown to be related to the severity of sleep alterations, decreased fronto-insular volumes and executive impairments in patients without any history of AW neurological complications [74]. Reasoning by analogy, we can think of the neuroprogression theory in affective disorders. This theory states that the repetition of depressive episodes, each one associated with transient cognitive deficits such as trouble concentrating and some level of invisible brain suffering, progresses toward permanent cognitive dysfunction, and later observable brain damages especially hippocampic destruction, but also larger grey matter reduction [75,76,77,78]. The same reasoning applies to the neurological consequences of traumatic brain injuries. A single traumatic brain injury, even mild, can induce neuroinflammation, metabolic abnormalities, oxidative stress and axonal damage. Repeatedly, it contributes to the development of chronic traumatic encephalopathy, a long-term neurodegenerative disease leading to dementia [79]. Although even a single traumatic brain injury can cause long-term neurological consequences, it has been shown that the progression of the disease and the degree of neuronal damage are correlated with head trauma repetition [80,81].

The neuroprogression theory applies to AW leading to persistent cognitive deficits. But at the end of the evolution when a severe cognitive disorder is diagnosed in a patient with a past history of severe AUD, despite comprehensive etiological inquiry [82], it is impossible to differentiate the proper impact of repetitive AW from the neurotoxic effect of alcohol use. The effects of alcohol intake and AW may be cumulative and also interact with other independent pathophysiological processes: the development of neurodegenerative disorders, a vascular dysfunction, events such as seizures or strokes, nutritional depletion, etc., lowering the threshold of significant impairment and advancing the diagnosis. This could explain why AUD is the most frequent risk factor of early-onset dementia and the most frequent modifiable risk factor of all dementia, including Alzheimer’s disease [6].

## 6. Perspective

Reducing the burden of AUD in terms of alcohol-related cognitive deficits should be a health priority. To be able to propose effective neuroprotective strategies for patients with AUD, especially during AW episodes, several steps forward in research are needed. The first step is reaffirming the relevance of thiamine prescription to avoid the development of WE, especially during the critical period of AW and potentially to reduce the prevalence of alcohol-related cognitive impairments. In parallel, it is crucial to work on raising public policy awareness for the reimbursement of thiamine.

The second one is to detect patients at risk to develop AW complications using new biomarkers. The validation of accessible, repeatable, and of course sensitive biomarkers of oxidative stress and brain suffering during AW is essential. If we want to move forward a precision medicine, we need to develop peripheral biomarkers of neuronal, axonal, synaptic but also astrocytic destruction applied to AW [83].

The third and final step is the development of innovative treatments to counteract the oxidative stress and prevent the neurotoxic effect of AW and its various complications.

## 7. Conclusions

There are now several pieces of evidence showing that AW is an oxidative stress challenge for the brain, leading to epilepsy and DT, and making this timeframe a vulnerability period for the development of WE, especially for AUD patients with subclinical thiamine deficiency. Reasoning by analogy, we support the opinion that this oxidative stress challenge induces brain lesions, which may lead through the repetition of episode through a neuroprogression effect toward an additive brain and cognitive loss. Future research should now focus on identifying and assessing this neurotoxic effect to develop specific neuroprotective strategies.

## Figures and Tables

**Figure 1 antioxidants-11-02078-f001:**
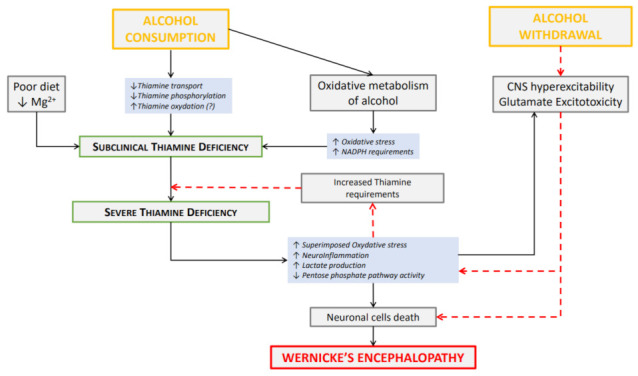
Relationships between alcohol consumption, alcohol withdrawal and Wernicke’s encephalopathy. Alcohol consumption may be responsible for subclinical thiamine deficiency due to elevated thiamine requirements for energy metabolism and oxidative alcohol metabolism, impaired thiamine distribution and metabolism, and impaired nutritional status. Red dotted arrows indicate the effect of alcohol withdrawal that induces central nervous system (CNS) hyperexcitability, glutamate excitotoxicity, and exacerbates oxidative stress, neuroinflammation and thiamine requirements leading in some cases to severe thiamine deficiency, synchronous neuronal cell death, including localized thalamic and mamillary bodies neuron death responsible for Wernicke’s encephalopathy. The «(?)» corresponds to a hypothesis.

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
