# Peer review of "Alcohol Withdrawal Is an Oxidative Stress Challenge for the Brain: Does It Pave the Way toward Severe Alcohol-Related Cognitive Impairment?"

_antioxidants, 2022, doi:10.3390/antiox11102078_

Round 1

Reviewer 1 Report

This is a review by Dr. Clergue-Duval and collaborators. The authors explore the hypothesis that alcohol withdrawal is an oxidative stress challenge for the brain.

Major Concerns

The authors did not discuss other mechanisms contributing to neuronal hyperexcitability following alcohol withdrawal, such as the upregulation of voltage-gated calcium channels and the down-regulation of calcium-activated potassium channels. The authors did not further discuss the sources (in addition to glutamate receptors) of elevated intracellular calcium following alcohol withdrawal. Limited evidence demonstrates that altered oxidative stress can trigger alcohol withdrawal seizures.

Minor concerns

Line 114. Carvalho et al., 2019. Please use 10

Line 130. Benzodiazepines are the first rank treatment. First choice? 

Line 178.  Seizures are provoked by the pre-existing hyperexcitability of neurons. What does pre-existing mean? Please clarify.

Lines 301-302. Hyperglutamatergia might not be the only mechanism leading to increased intracellular calcium.

Reviewer 2 Report

The review article by Clergue-Duval et al., introduce and elucidate an  
important issue about
alcohol-related neurocognitive disorders caused by chronic and  
excessive alcohol use.
The authors explain and suggest a potential development of  
neuroprotective strategies during alcohol withdrawal in patients with  
severe alcohol use disorders (AUD).
They sum up recent progress in the field of alcohol use disorders and  
neurological effects.
The literature cited is of important relevance, and the author is  
aiming to introduce their own style, interpretation as well as and  
outlook on this interesting topic. Interestingly, the author arranges  
a review that does hold an individual flavor.
In addition, the figure is captivating, both for the layout and for  
the information it intends to provide. In order to arouse the interest  
of the reader, a table and/or box is proposed which contains  
concluding remarks and open questions. Alternatively, a graphical  
abstract is suggested. This is a well-written review and the length of  
the article is commensurate with the message.
For the mentioned reasons, the manuscript does not need originality  
resulting in a review that guides the reader stringently through the  
story. The manuscript may be accepted for publication with minor  
revision.

Reviewer 3 Report

This review is an interesting insight, and it could be a new illustration of the double-edged sword role of oxidative stress about Alcohol use disorders and Alcohol withdrawal which can cause the onset of neurodegenerative disease by disturbing neural networks due to oxidative stress-related cellular events and molecular reprogramming with aging. So, I agree with the author’s opinion that a potential model of brain disorders like AD and related diseases goes the further value of the exploration without any questions.